# Clinical and economic burden of varicella in pediatric patients hospitalized in four institutions in Guatemala

**Mario Melgar**[1,2,3], **Ingrid Sajmolo**[1,4,*], **André Chocó**[2,5], **Lidia Ortiz**[1],
**Ana Gabriela Cordova**[2], **Luis Hernández**[6], **Irwing Rivera**[2], **Ashly Zuñiga**[1], **Claudia Beltrán**[7],
**Sebastian Medina**[7], **Marcel Marcano-Lozada**[8]

**1** Centro de Estudios Clínicos Salud Avanzada, Guatemala, Guatemala, **2** Hospital Roosevelt, Guatemala, Guatemala, **3** Unidad Nacional de Oncología Pediátrica, Guatemala, Guatemala, **4** Hospital General San Juan de Dios, Guatemala, Guatemala, **5** Unidad de Investigación, Clínica de Atención Integral VIH, Hospital Roosevelt, Guatemala, Guatemala, **6** Hospital Infantil de Enfermedades Infecciosas y Rehabilitación, Guatemala, Guatemala, **7** MSD LATAM, Bogotá, Colombia, **8** MSD LATAM, San José, Costa Rica

\* sajmolo30@gmail.com

## Abstract

Varicella presents a public health challenge in Guatemala, with limited evidence regarding its impact; vaccine is currently absent from the national immunization program. Generating local data on the economic and health burden can support immunization policies. This study describes the use of hospital resources, costs of care, clinical and demographic characteristics, and complications in children with varicella. A retrospective review of medical records from patients admitted in four public institutions in Guatemala between January 2015 and December 2019, with diagnosis of varicella was performed. A total of 124 hospitalized patients, aged 0 to 14 years, were analyzed, of whom 46% had cancer. The average cost of hospitalization was USD 3,793.24 for complicated cases and USD 1,131.11 for uncomplicated cases. The average hospital stay was 11 days for patients with complications and 5 days for those without complications. In total, 66.1% of patients presented complications, with related infection complications being the most common. In conclusion, hospitalizations for varicella impose a significant economic and clinical burden in Guatemala. Complications increase the cost of care by three times, highlighting the importance of considering the inclusion of the varicella vaccine in the National Immunization Program to prevent the disease.

## Introduction

Varicella is an infectious disease caused by the varicella-zoster virus (VZV). It is highly contagious and usually occurs during childhood [1]. Various factors such as age, immunosuppression, and climate influence its incidence [2]. Although it generally has a benign course, it can also result in serious complications like bacterial superinfection of skin lesion with or without bacterial sepsis, central nervous system involvement, pneumonia, other complications are myocarditis, corneal lesions, nephritis, arthritis, acute glomerulonephritis, hepatitis, and

**Data availability statement:** All relevant data are available from figshare at https://doi.org/10.6084/m9.figshare.28057484.

**Funding:** This study was funded by Merck Sharp & Dohme (MSD) in the form of grant [NIS009952] to MM via Centro de Estudios Clínicos en Salud Avanzada (CECLISA) and in the form of salaries to CB, SM, and MM-L. The specific roles of these authors are articulated in the "author contributions" section. The funders were involved in the protocol design and manuscript preparation, but were not involved in data collection, analysis, or the decision to publish.

**Competing interests:** The authors have read the journal's policy and have the following competing interests: Mario Melgar has received research grants from Merck Sharp & Dohme (MSD) and Pfizer, and has participated in an expert forum for MSD. Ingrid Sajmolo received a travel grant from MSD. Claudia Beltran, Sebastian Medina, and Marcel Mecano-Lozada are employees of MSD. There are no patents, products in development or marketed products associated with this research to declare. This does not alter our adherence to PLOS ONE policies on sharing data and materials.

thrombocytopenia [1,3]. Giglio et al. (2018) reported that up to 28% of outpatients and 98.7% of hospitalized patients experienced one or more complications [4]. This study was conducted in Guatemala, an upper-middle-income country located in Central America. At the time of the study, the varicella vaccine was not included in the National Immunization Program, but it was available at no cost to people with access to Social Security and could be purchased through private practice. Varicella is not notifiable in the health system, which limits the availability of information on its incidence, behavior, and complications [5]. The objective of this study is to describe the clinical characteristics, complications, health resource utilization (HCRU), and costs associated with health care in pediatric patients hospitalized for varicella.

## Materials and methods

### Ethics statement

The protocol was submitted in accordance with the local requirements of each site. Initially, version 1.0 of the protocol was submitted to the independent Ethics Committee of the Hospital Roosevelt (HR). Subsequently, it was submitted and approved by the research committee of the Hospital General San Juan de Dios (HG), the Teaching Committee of the Hospital Infantil de Infectología y Rehabilitacion (HI), and the Academic Committee of the Unidad Nacional de Oncologia Pediatrica (UNOP), prior to the start of data collection from all sites. We confirm that the study was reviewed by the Independent Ethics Committee, along with the respective research committees, which determined that informed consent was not required for this retrospective study of medical records. All information was collected anonymously by treating physicians or research staff, and study patients were identified only by an encrypted patient number. The authors did not have access to information that could identify individual participants during or after data collection. Approval was obtained from each site's local independent research and ethics committee prior to commencing data collection, which started on 21/04/2022. The first subject was enrolled on 30/05/2022 in the UNOP. Subsequent enrollments were as follows: 08/09/2022 in the HG; 23/06/2022 in the HI; and 26/07/2022 in the HR.

This was a multicenter study, with a review of medical records, conducted in four hospitals located in Guatemala City: three national reference general hospitals and one pediatric oncology hospital. There is no official proportion of the Guatemalan population that these hospitals cover. However, since they are the only reference hospitals in the area, they attend to most of the complicated varicella cases from the metropolitan area and some from other parts of the country. Medical records of patients aged 0 to 14 years were reviewed, whose clinical records were available, and who were hospitalized from January 1, 2015, to December 31, 2019, with a primary discharge diagnosis of varicella (ICD-10 Code: B01) and/or any complications related to varicella according to clinical examination during the study period.

For each patient, the number of workdays lost by the caregiver was determined based on the duration of hospitalization, disability, or, if this information was not available, 2.5 days who were considered according to the literature [6,7]. The costs incurred when they were treated on an outpatient basis before admission to the hospital were also reviewed.

Direct costs were calculated using data from national forms, insurance, and official national sources. All costs were expressed in local currency (quetzales), and an average exchange rate from the national bank for the entire period from 2015 to 2019 was applied (1 USD = 7.579926 quetzales). Costs prior to the study period were adjusted for inflation. For the year 2022, the reported exchange rate was 7.76808 quetzales per 1 USD. Additionally, all costs were adjusted for cumulative inflation up to 2022 to reflect their updated value. To calculate the cost of a daily workday, data on the national average minimum wage of USD 11.91 (Q. 93.55), as reported by the Ministry of Labor and Social Security of Guatemala in 2022, were used [8].

Descriptive statistics were used to summarize costs associated with health care and sociode-mographic and clinical information. For categorical variables, measures of observed frequency and relative frequency (percentage) were calculated. For quantitative variables, measures of central tendency and variability were calculated (mean, standard deviation [SD], median, interquartile range [IQR], percentiles).

## Results

In total, the medical records of 124 hospitalized patients were analyzed. These were distributed as follows: 16 patients in HG, 18 in HI, 33 in HR, and 57 in UNOP. Of these patients, 41.1% (51) were female and 58.9% (73) were male. The average age of hospitalized patients was 4.5 years (SD 3.3), with children aged 3 to 6 years being the most affected. 57.3% belonged to urban areas, and 42.7% to rural areas. 60.5% had a normal nutritional status, 17.7% were malnourished, 10.5% were overweight, and 11.3% were obese. A total of 46% (57) had cancer, 7.3% (9) suffered from malnutrition, HIV, rheumatic disease/steroid use respectively, or another predisposing condition in 1.6% (Table 1). In 73.4% (91) of the patients, the number of skin lesions caused by varicella was unknown, and 22.6% had fewer than 50 lesions. Only 10.5% (13) of the patients had known contact with another person with varicella; none of the patients had contact with cases of herpes zoster; in the majority, this was not stated in the medical record. 21.8% of patients required admission to intensive care units. 98.4% of patients were hospitalized only once, while 1.6% were hospitalized twice.

Complications occurred in 82 cases (66.1%). In patients with complications, the mean age was 4.6 years (SD 4.3); 61% were male, and 39% were female. Among patients with complications, 32.9% also had cancer. Regarding the number of complications, 53.2% had one or two complications, and 12.9% had three or more complications. The most frequent complications were related with infections, mainly skin and soft tissue infections (34.7%), followed by pneumonia (13.7%) and encephalitis (8.9%). The most frequent non-infectious complications were hematological alterations (8.1%) and dehydration (4.0%). Of the patients with complications, 32.9% required admission to the ICU; none of the patients without complications required admission to the ICU.

74.2% of the patients recovered, 21.8% recovered with sequelae, and 2.4% died. Of the patients recovered with sequelae, 13.7% (17) presented skin and soft tissue sequelae, 5.6% (7) presented neurological sequelae, and 2.4% (3) had other sequelae. Complications were more frequent in those who recovered with sequelae, and the three patients who died had complications from varicella, which were pneumonia, soft tissue infection, and sepsis. Of the patients who died, two were not immunocompromised, and one patient had cancer.

For HCRUs, an average of 10.6 laboratory tests and 1 image per patient were utilized, the most common was the X-rays. 55.6% of the patients used over-the-counter medications, 66.1% used antibiotics, and 69.4% used medications to manage the disease such as analgesics, immunoglobulins, among others. The use of antibiotics varied among the patients studied. It was observed that 31% (13) of patients without complications and 84.1% (69) of patients with complications received antibiotics. The average length of hospital stay was 9.4 days (SD 10.4), and the number of days in the ICU was 1.6 days (SD 3.8) (Tables 1 and 2).

The average length of hospital stay for patients with complications was 11 days (SD 12), compared to 3 days (SD 9) for patients without complications. The following unit costs were identified: the medical consultation or outpatient visit had a cost of USD 21.97, according to data obtained from local insurance companies. From publicly available information and published rates, the following costs were derived: hospital stay per day, USD 144.86; stay in the intensive care unit (ICU), USD 566.41 per day, as reported by Alvis-Guzmán et al. (2018) [9]; USD 8.40 for professional advice in consultation with other medical specialties, as established

**Table 1. Demographic and epidemiological characteristics of hospitalized patients with varicella in four institutions in Guatemala City, 2015 to 2019, n = 124.**

| | Complications | | | | | |
| --- | --- | --- | --- | --- | --- | --- |
| | Yes | | No | | Total | |
| | ƒ | % | ƒ | % | ƒ | % |
| Hospital | | | | | | |
| HG | 15 | 18.3% | 1 | 2.4% | 16 | 12.9% |
| HI | 11 | 13.4% | 7 | 16.7% | 18 | 14.5% |
| HR | 28 | 34.1% | 5 | 11.9% | 33 | 26.6% |
| UNOP | 28 | 34.1% | 29 | 69.0% | 57 | 46.0% |
| Age (years) | | | | | | |
| < 3 | 27 | 32.9% | 15 | 35.7% | 42 | 33.9% |
| 3 a 6 | 32 | 39.0% | 17 | 40.5% | 49 | 39.5% |
| 7 a 10 | 19 | 23.2% | 7 | 16.7% | 26 | 21.0% |
| 11 a 14 | 4 | 4.9% | 3 | 7.1% | 7 | 5.6% |
| Mean, SD | 4.6 | 4.3 | 3.4 | 3.2 | 4.5 | 3.3 |
| Gender | | | | | | |
| Female | 32 | 39.0% | 19 | 45.2% | 51 | 41.1% |
| Male | 50 | 61.0% | 23 | 0.547619 | 73 | 58.9% |
| Ethnicity | | | | | | |
| Ladino | 66 | 80.5% | 33 | 0.785714 | 99 | 79.8% |
| Maya | 16 | 19.5% | 9 | 21.4% | 25 | 20.2% |
| Area or residence | | | | | | |
| Urban | 44 | 53.7% | 27 | 64.3% | 71 | 57.3% |
| Rural | 38 | 46.3% | 15 | 35.7% | 53 | 42.7% |
| Nutritional condition | | | | | | |
| Malnutrition | 16 | 19.5% | 6 | 14.3% | 22 | 17.7% |
| Normal | 49 | 59.8% | 26 | 61.9% | 75 | 60.5% |
| Overweight | 7 | 8.5% | 6 | 14.3% | 13 | 10.5% |
| Obesity | 10 | 12.2% | 4 | 9.5% | 14 | 11.3% |
| Inmunocompromised | | | | | | |
| None | 49 | 59.8% | 8 | 19.0% | 57 | 46.0% |
| Cancer | 27 | 32.9% | 30 | 71.4% | 57 | 46.0% |
| Malnutrition | 6 | 7.3% | 3 | 7.1% | 9 | 7.3% |
| HIV | 0 | 0.0% | 2 | 4.8% | 2 | 1.6% |
| Rheumatic disease/ steroid use | 1 | 1.2% | 1 | 2.4% | 2 | 1.6% |
| Others | 2 | 2.4% | 0 | 0.0% | 2 | 1.6% |

SD: Standard deviation.

by the Ministry of Public Health and Social Assistance of Guatemala for the year 2022 [10] Haga clic o pulse aquí para escribir texto.. Table 3 describes the HCRU of the total 124 patients, including the types of specialists involved and the number of evaluations by other specialists. The total number of visits from one specialist was not taken into account. Outpatient visits before hospitalization (extracted from the medical history provided by the patients and recorded) and hospitalization bed-days (not including ICU) are also described.

The care and associated costs are shown in Fig 1 and Table 4. It was found that the average total cost of care was USD 2891.55 (SD 4179.35); USD 3793.24 (SD 4,878.20) for patients with complications, and USD 1,131.11 (SD 766.02) for patients without complications (Fig 1).

**Table 2. Average use of health resources per patient associated with varicella in four institutions in Guatemala City, 2015 to 2019, n = 124.**

|  | *Mean* | *N* | **% of patients** |
|---|---|---|---|
| Laboratory tests | 10.7 | 122/124 | 98.4 |
| Cultures | 1 | 68/124 | 54.8 |
| Imaging tests | 1 | 53/124 | 42.7 |
| Surgeries | 0.1 | 13/124 | 10.5 |
| Procedures | 0.5 | 29/124 | 23.4 |
| Over the counter medications (OTC) | 0.9 | 69/124 | 55.6 |
| Prescription medications | 0.8 | 105/124 | 84.7 |
| Antibiotics | 1.5 | 82/124 | 66.1 |
| Antivirals | 0.7 | 86/124 | 69.4 |
| ICU stay (days) | 1.6 | 27/124 | 21.8 |
| General ward (days) | 7.8 | 121/124 | 97.6 |
| Total days of hospital stay | 9.4 | 124/124 | 100 |

For the hospitalization and intensive care cost data, the inflation values reported by the Bank of Guatemala were used to ensure consistency with the other costs of medications, antivirals, antibiotics, procedures, and surgeries. The categories with the highest average costs in patients with complications were general care (USD 1,309.08), intensive care (USD 1,305.51), antibiotics (USD 317.31), prescription medications (USD 265.92), and laboratory tests (USD 151.73). The categories with the highest average costs for patients without complications were bed-days (USD 748.46), prescription medications (USD 116.02), antivirals (USD 62.87), and laboratory tests (USD 47.98).

In comparing patients with varicella who developed complications to those who did not, significant associations were observed with hospital location and immunocompromised status. Patients treated at certain hospitals exhibited higher rates of complications ($\chi^2$ = 18.23, $p < 0.001$), suggesting variability in outcomes across facilities. Immunocompromised patients, particularly those with cancer, were significantly more likely to experience complications ($\chi^2$ = 18.36, $p < 0.001$). No significant differences were found between the groups regarding age, gender, ethnicity, area of residence, or nutritional status, indicating these factors did not significantly influence the likelihood of complications in this cohort.

When comparing patients with varicella who developed complications to those who did not, significant differences were observed in healthcare utilization patterns. Patients with complications were more likely to consult a specialist, with 62.3% doing so compared to 31.0% of non-complicated cases ($\chi^2$ = 11.64, $p < 0.001$). They also had a higher number of specialist consultations, with 24.4% of complicated cases receiving four or more consultations, whereas none were reported in the non-complicated group ($\chi^2$ = 13.70, $p < 0.001$). Additionally, patients with complications had more outpatient visits, with 41.5% having at least one visit compared to 19.0% of non-complicated patients ($\chi^2$ = 6.31, $p = 0.012$), indicating a greater demand for medical attention among those who developed complications.

Independent samples Student's t-tests were conducted to compare the length of hospital stay between varicella patients who developed complications and those who did not. Patients with complications had a significantly longer general ward stay (Mean = 9 days, SD = 10) compared to non-complicated patients (Mean = 5 days, SD = 3), t(106) = 3.34, $p = 0.001$. Similarly, the total bedridden stay was longer for complicated cases (Mean = 11 days, SD = 12) versus non-complicated cases (Mean = 5 days, SD = 3), t(99) = 4.28, $p < 0.001$. A t-test could not be performed for ICU length of stay because the non-complicated group showed no

**Table 3. Health resource utilization of patients hospitalized with Varicella in four institutions in Guatemala City, 2015 to 2019, n = 124.**

| | Complicated | | Non-complicated | | Total | |
|---|---|---|---|---|---|---|
| | N | % | N | % | N | % |
| Specialist Consultations | | | | | | |
| Infectologists | 14 | 17.1% | 6 | 14.3% | 20 | 16.1% |
| Oncologists | 14 | 17.1% | 7 | 16.7% | 21 | 16.9% |
| Pediatric Surgeon | 7 | 8.5% | 0 | 0.0% | 7 | 5.6% |
| Intensivist | 6 | 7.3% | 0 | 0.0% | 6 | 4.8% |
| Neurologist | 4 | 4.9% | 0 | 0.0% | 4 | 3.2% |
| Traumatologist | 4 | 4.9% | 0 | 0.0% | 4 | 3.2% |
| Dermatologist | 3 | 3.7% | 0 | 0.0% | 3 | 2.4% |
| Nutritionist | 3 | 3.7% | 0 | 0.0% | 3 | 2.4% |
| Cardiologist | 3 | 3.7% | 0 | 0.0% | 3 | 2.4% |
| Plastic Surgeon | 3 | 3.7% | 0 | 0.0% | 3 | 2.4% |
| Psychiatrist | 2 | 2.4% | 0 | 0.0% | 2 | 1.6% |
| Pulmonologists | 2 | 2.4% | 0 | 0.0% | 2 | 1.6% |
| Others | 2 | 2.4% | 0 | 0.0% | 2 | 1.6% |
| None | 43 | 52.4% | 30 | 71.4% | 73 | 58.9% |
| Number of specialist consultations | | | | | | |
| None | 20 | 24.4% | 9 | 21.4% | 29 | 23.4% |
| One to three | 42 | 51.2% | 33 | 78.6% | 75 | 60.5% |
| Four or more | 20 | 24.4% | 0 | 0.0% | 20 | 16.1% |
| Outpatient visits by hospitalized patients | | | | | | |
| Emergency room | 24 | 29.3% | 5 | 11.9% | 29 | 23.4% |
| Private clinic physician | 15 | 18.3% | 4 | 9.5% | 19 | 15.3% |
| Clinic | 14 | 17.1% | 2 | 4.8% | 16 | 12.9% |
| Hospital outpatient visit | 10 | 12.2% | 2 | 4.8% | 12 | 9.7% |
| Visits to health centers | 6 | 7.3% | 1 | 2.4% | 7 | 5.6% |
| Number of outpatient visits | | | | | | |
| None | 48 | 58.5% | 34 | 81.0% | 82 | 66.1% |
| One | 16 | 19.5% | 3 | 7.1% | 19 | 15.3% |
| Two | 6 | 7.3% | 3 | 7.1% | 9 | 7.3% |
| Three | 6 | 7.3% | 2 | 4.8% | 8 | 6.5% |
| Four or more | 6 | 7.3% | 0 | 0.0% | 6 | 4.8% |
| General Ward stay (days) | | | | | | |
| Media, SD | 9 | 10 | 5 | 3 | 8 | 8 |
| Length of stay in ICU (days) | | | | | | |
| Media, SD | 2 | 4 | 0 | 0 | 2 | 4 |
| Total hospital stay (days) | | | | | | |
| Media, SD | 11 | 12 | 5 | 3 | 9 | 10 |

variation (Mean = 0 days, SD = 0). These results indicate that developing complications is associated with a longer hospital stay in varicella patients.

Complications were less frequent in patients treated at the oncology institution than in other hospitals, and these complications were less frequent in immunocompetent patients. A detailed cost analysis was conducted for patients with and without cancer, and with or without complications. It was found that the average cost for non-oncological patients with

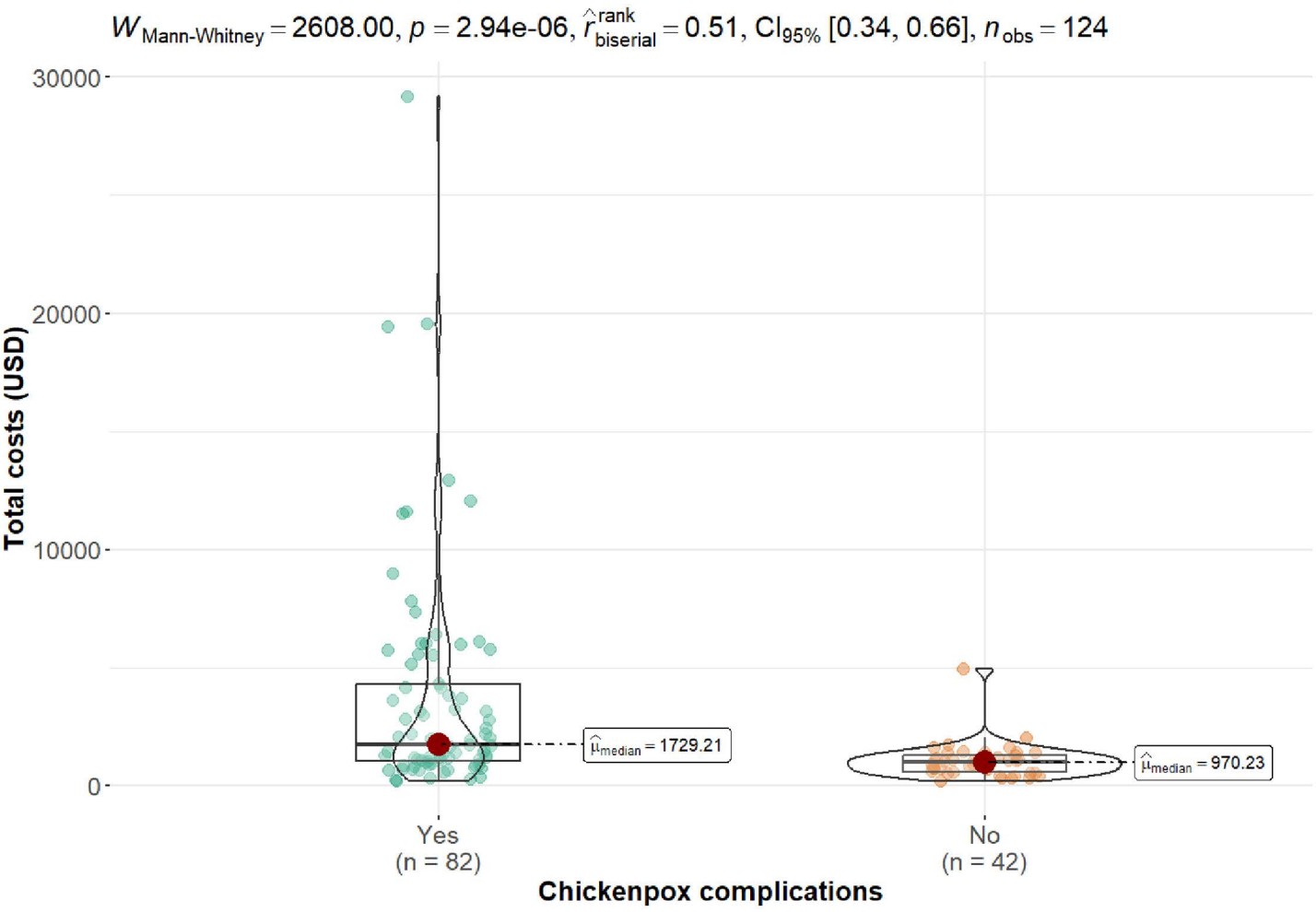

**Fig 1. Comparison of total healthcare costs for hospitalized varicella patients with complications.**

complications was USD 3,872.62, while for oncological patients with complications it was USD 3,640.14. For uncomplicated and healthy patients, the cost was USD 1,217.05, and for uncomplicated cancer patients it was USD 1,092.58. These differences between groups were statistically significant with a p value < 0.05.

Finally, the costs of hospitalization for varicella in patients with and without complications were compared. It was found that indirect costs were significantly higher in patients with complications, with an average cost of USD 3,793.24 (SD 4,878.20) for patients with complications and USD 1,131.11 (SD 766.02) for patients without complications. A marked difference in the cost of treatment between hospitalized patients with varicella complications and those without complications was observed, with the cost being 3.35 times higher for the former (p < 0.001).

## Discussion

The study focused on describing the clinical characteristics, complications, HCRU utilization, and costs associated with the care of pediatric patients hospitalized for varicella in Guatemala. It was found that varicella in this population implies a significant use of health and economic resources. This analysis was based on inpatient data, excluding outpatient cases for comparative purposes.

**Table 4. Comparison of costs associated with health care for patients hospitalized with varicella according to complications in four institutions in Guatemala City, 2015 to 2019, n = 124.**

| Type of cost | Complicated (n = 82) | Non-complicated (n = 42) | Total (n = 124) |
|---|---|---|---|
| | Average cost in USD (SD) | Average cost in USD (SD) | Average cost in USD (SD) |
| Outpatient visit costs | | | |
| Laboratory tests | 0.55 (3.25) | 0.87 (3.93) | 0.66 (3.48) |
| Imaging tests | 0.65 (5.85) | 1.26 (5.71) | 0.85 (5.79) |
| Outpatient consultations | 12.59 (21.29) | 4.18 (11.10) | 9.74 (18.85) |
| Medications | 1.25 (6.71) | 0.75 (3.05) | 1.08 (5.73) |
| Antibiotics | 0.08 (0.54) | 0.00 (0.00) | 0.05 (0.44) |
| **Total outpatient costs** | 15.12 (24.99) | 7.06 (16.86) | 12.39 (22.82) |
| Hospitalization costs | | | |
| General ward stay | 1309.08 (1434.34) | 748.46 (370.25) | 1119.19 (1213.05) |
| Intensive care unit stay | 1305.51 (2509.02) | 0.00 (0.00) | 863.32 (2128.49) |
| Surgery | 23.42 (81.61) | 0.00 (0.00) | 15.48 (67.16) |
| Laboratory tests | 151.73 (376.85) | 47.98 (58.26) | 116.59 (311.58) |
| Cultures | 33.13 (34.45) | 7.04 (11.62) | 24.29 (31.31) |
| Consultations with other specialists | 21.41 (25.05) | 11.00 (7.30) | 17.88 (21.34) |
| Procedures | 104.43 (315.61) | 0.55 (3.54) | 69.24 (260.84) |
| Imaging tests | 64.15 (106.96) | 1.89 (6.90) | 43.06 (91.78) |
| Prescribed medications | 265.92 (792.75) | 116.02 (667.27) | 215.15 (753.22) |
| Antibiotics | 317.31 (879.63) | 22.29 (46.22) | 217.38 (727.95) |
| Antivirals | 60.49 (65.65) | 62.87 (47.87) | 61.30 (60.03) |
| Over-the-counter medications | 11.35 (16.45) | 7.83 (11.18) | 10.16 (14.92) |
| **Total hospitalization costs** | 3641.06 (4803.80) | 1033.18 (752.81) | 2757.75 (4113.56) |
| **Direct costs** | 3655.80 (4804.00) | 1039.77 (755.13) | 2769.73 (4115.02) |
| **Indirect costs** | | | |
| Cost of caregiver days lost from work* | 137.44 (117.96) | 91.34 (30.45) | 121.82 (99.76) |
| **Total** | 3793.24 (4878.20) | 1131.11 (766.02) | 2891.55 (4179.35) |

SD = Standard deviation.

*The assumption of 2.5 workdays lost was implemented in the analysis as part of the calculation for indirect costs, it was an assumption based on previously published literature, not a value derived from patient-specific data.

It is important to note that treatment protocols vary between institutions. At UNOP, most patients are hospitalized even with a less severe form of the disease due to their oncological condition. In this institution, patients are admitted even when they do not present complications, whereas other hospitals only admit patients with infectious and other complications. This variation in treatment protocols could affect costs between institutions. It is possible that some costs are underestimated, and that only certain resources used are documented in clinical records. However, the data presented provide an important view of the country's costs of the disease and HCRU.

In this study, the average hospital stay (10 days) was longer compared with data from the study conducted by Wolfson et al. (2018), with a notable percentage of patients requiring intensive care and a prolonged ICU stay. In the study conducted by Wolfson et al. (2018) in five middle-income countries, the longest hospital stay was in Peru at 6.7 days. Regarding ICU admission, the longest stays occurred in Mexico and Peru, lasting 6.7 days; in Mexico, 19.5% of patients required ICU admission [11].

The length of hospital stay could be longer due to complications, but compared to reports from Peru and Argentina, complication rates were lower. Another reason for the prolonged hospital stay could be the profile of the patients, as a large proportion had comorbidities, especially cancer. These data indicate that, although the percentage of patients required ICU admission is lower, the average hospital stay in pediatric intensive care (7.5 days for 21.8% of patients) is longer compared to countries such as Argentina, which has an average of less than 4.8 days, and Peru, with an average of 7.0 days [4,11,12].

Complications related to varicella were recorded in more than half of the patients, and three fatal cases (2.4%) were documented. These deaths indicate a high risk of mortality due to complications derived from varicella: two were caused directly by the virus (pneumonia and sepsis), and the third was a consequence of secondary complications (soft tissue infection). Although the proportion of deaths was low, all cases occurred in patients who presented complications. No significant differences were found in terms of nutritional status, age, gender, and ethnicity when comparing the groups with and without complications. The medical records of patients with varicella were examined to investigate the relationship between the number of skin lesions during the rash and the frequency of complications. Despite not finding evidence of a higher incidence of complications depending on the number of lesions, the limited availability of data in the medical records hindered any conclusive analysis. Therefore, more detailed data recording is required to establish a more precise association. It is important to note that varicella and its complications are not notifiable in the country and are not a priority disease in terms of epidemiological surveillance.

In contrast to Argentina and Peru, deaths related to varicella were recorded in Guatemala. In a systematic review of the burden of varicella in Latin America, Arlant et al. (2019) report that before the introduction of vaccination, there was limited data on mortality associated with varicella. Information is only available from three studies carried out in Brazil, Colombia, and Costa Rica. For example, Brazil reported an average annual mortality from varicella of 0.88 cases per 100,000 inhabitants in children under 1 year of age and 0.02 cases per 100,000 inhabitants in the 15 to 19 age group, with a total of 2,334 deaths associated with varicella between 1996 and 2011 [13]. In the study by Pawaskar et al. (2022), a time series model was used with surveillance data in Colombia. This analysis compared mortality before (2008–2015) and after (2015–2019) the implementation of universal varicella vaccination. The results showed a reduction in mortality in the general population, decreasing from 0.8 to 0.5 per 1,000,000 inhabitants, and from 1.3 to 0.5 per 1,000,000 in children aged 1 to 4 years old [14].

Because notification of varicella cases is not mandatory in Guatemala, the incidence and utilization of outpatient and hospital resources related to the disease burden may be underestimated. In this study, the use of outpatient resources by patients who required hospitalization was analyzed. Although outpatient costs were found to be minimal compared to hospital costs, this figure could be an underestimate because most of the treatment was performed during hospitalization.

In Guatemala regarding laboratory studies, blood cultures were the most common, an average of one imaging study per patient was found, with x-rays being the most frequently performed. Compared to other Latin American countries, the study by Wolfson et al. (2019) reports a greater number of laboratory tests. Although in countries such as Mexico and Hungary more cultures and imaging studies are performed per patient, the data presented here show lower averages, even though x-rays remained the most common in all countries [11].

For prescribed medications in Guatemala, patients used an average of 1.6 over-the-counter medications and received 2.3 antibiotics. Additionally, 84.2% of patients were taking some medication, with an average of 2.1 medications per patient. This is similar to what was

reported in Poland, where 1.5 antibiotic prescriptions per patient were recorded. However, the average number of antibiotics per patient in Guatemala is slightly lower than the 2.5 antibiotics per patient used in Mexico. In comparison, 94% of patients in Peru and Poland were reported to be taking at least one medication, while in Argentina this figure was 77.3% [4,11].

Regarding costs, the expense per hospitalized patient is Guatemala is lower than in Mexico, where the cost is 5,786.20 USD per patient. However, Guatemala's cost is higher compared to other Latin American countries, such as Argentina (2,947.70 USD) and Peru (769.90 USD) per hospitalized patient. These costs are significantly lower than the HCRU observed in European countries such as Hungary and Poland, as reported in the study by Wolfson et al. (2019), which identified these countries as having the highest costs. These costs will be important to consider when evaluating the potential inclusion of the varicella vaccine in the national immunization program [11].

In comparison with other Latin American literature, the complication rate was lower in patients in Guatemala. Among patients who required surgery for complications, the most common procedures were abscess drainage, lavage and debridement, and placement of continuous negative pressure devices. These types of procedures increase costs in patients with varicella complications. An in-depth analysis compared patients with and without HRU (Health Resource Utilization) complications and found that the costs for patients with complications were 3.5 times higher than those for patients without complications. Except for UNOP, the other centers hospitalized a higher proportion of complicated patients, and the cost of treating these patients was higher than treating uncomplicated patients. This implies that varicella represents a significant disease burden for the country and the national health system. As seen in Table 4, complicated patients have a greater variation in care costs. Costs for cancer patients vary due to differences in the standard of care. A group analysis was performed to determine the difference between cancer patients with and without complications versus immunocompetent patients with and without complications. It was found that the cost for cancer patients without complications is lower. O'Connor et al. (2014) reported a mortality rate of 12% due to viral infections, including varicella, in pediatric patients with acute lymphoblastic leukemia [15]. A previous study conducted at UNOP from January 2009 to March 2013 found that the estimated minimum average cost of an episode of varicella in children treated for cancer was 598.75 USD, considering an average stay of 6 days. Our findings show that currently, the cost is double what was previously found [16].

It is essential to highlight the cost of lost workdays for caregivers, with an average value of USD 137.44 and USD 91.34 for patients with and without complications, respectively, which represents a 50% increase in the cost of lost workdays for caregivers. In Guatemala, the value of a workday during the study period was USD 11.91, which implies that if the caregiver works in the informal economy, hospitalization means at least 7.6 days of lost support for the family unit. This exposes caregivers to a high socioeconomic risk that accentuates the poverty of the household members.

There is no country-level data on the incidence of varicella and its complications for Guatemala, and for most Latin American countries, varicella is not a notifiable disease. However, according to conservative estimates, varicella is responsible for 4.2 million serious complications leading to hospitalization and 4,200 deaths worldwide each year [13]. In the pre-vaccination era, in high-income developed countries, the case fatality rate for varicella was 3 deaths per 100,000 cases [17]. In Costa Rica, during this period, more than 80% of cases occurred in children under 14 years of age, resulting in the hospitalization of 872 children over a 10-year period, of which 631 (72.4%) were under 12 years of age [18]. Annual hospitalization rates and similar fatalities are estimated in cases of natural varicella as in outbreaks,

with hospitalization rates of 0.297%, 0.129%, 0.147%, 0.269% for children ages under 1, 1 to 4, 5 to 9, and 10 to 14 years, respectively; and mortality rates of 0.0106%, 0.0027%, 0.0018%, 0.0031% for children under 1, 1 to 4, 5 to 9, and 10 to 14 years, respectively [19]. Using these data to estimate cases in Guatemala, and considering the INE (National Institute of Statistics Guatemala) population, it is estimated that by 2024 there will be a population of 4,809,312 children under 12 years of age. Extrapolating the data annually for the Guatemalan population, an estimated 30,000 cases (28,800–38,400) of patients with varicella are projected. A total of hospitalized children is expected by age, being 396, 567, 664, and 1,030 for the groups under 1, 1 to 4, 5 to 9, and 10 to 14 years, respectively, for a total of 2,657 children. According to studies in Brazil, 12-month-old children have a higher risk of complications (14.3%), while for other ages a lower percentage (5.7%) of complications is recorded [13]. Therefore, 57 children under one year of age and 148 children between 1 and 14 years of age are expected to present complications, for an estimated total of 205 affected. Annual costs are estimated to be USD 777,614.20 (SD 1,000,031), with a prognosis of 46 deaths.

The effectiveness of universal vaccination against varicella has been widely studied in various countries, demonstrating striking reductions in the incidence of complications and hospitalizations. For example, in Germany, after the introduction of routine vaccination, a 90% reduction in complications was observed in children under 5 years of age, which represented a significant milestone in the protection of children's health [20]. In Costa Rica, where the vaccine was implemented in 2007, an impressive 95% coverage was achieved by 2015, resulting in a 73.8% decrease in varicella cases and a massive 98% reduction in hospital complications [18]. Argentina, after the introduction of the vaccine in 2015, with coverage of 77.6% by 2019, also experienced a notable decrease of 83.9% in the incidence of the disease [21]. In Colombia, introduced into the vaccination scheme in 2015 with 90% coverage, a 70.5% reduction in incidence and a significant decrease in general mortality was evident [14]. Furthermore, experiences such as that of Uruguay, where a complication rate of 7% was found in vaccinated children compared to 12% in unvaccinated children [13], underline the positive impact of vaccination in protecting children's health throughout the region. These findings support universal vaccination as an effective strategy to reduce both the burden of disease and the costs associated with healthcare in Latin America.

Avila-Aguero et al. (2016) conducted a cost-benefit analysis in Costa Rica prior to the introduction of the varicella vaccine. They used the statistical tool Varicella Vaccination Tool for Economic Analysis (EVITA). With 85% coverage among 15-month-old children, it was estimated that after 3 years at least 44,000 cases and 9,000 serious complications could be prevented. This would generate savings of $17,000,000 in direct care services, with an annual investment of $1,125,000 in vaccines. The cost-benefit ratio was 41 for society and 30 for social security, which means that for every dollar invested, between 30 and 41 dollars were obtained in terms of medical care and lost work [18].

The study by Giglio et al. (2023) evaluated the economic impact of the universal varicella vaccine in Argentina, which was implemented in July 2015. Before vaccination, varicella treatment cost around $40 million for children aged 1 to 14 years. Since July 2015, significant reductions in costs were observed, reaching a total of -$65.4 million for the post-vaccination period (July 2015 to December 2019). These reductions affected both direct and indirect costs. The greatest reductions were seen in the years with the highest incidence of varicella, with 2019 being the most notable year in terms of reduction. Additionally, cost reductions were estimated in the general population, which went from $22.1 million in 2016 to $40.0 million in 2019. In specific terms per eligible child and per vaccinated child, the cost reductions were $28.21 and $36.31 respectively for the target population from 1 to 4 years old, and $53.61 and $69.00 respectively for the general population [21].

## Limitations

Inherent limitations of the study arise from its non-probability sampling design and retrospective chart review. Data for patients treated on an outpatient basis are not routinely recorded and therefore could not be collected. Some hospitalized patient data were not recorded and some medical records could not be located because the country has a paper registry and not an easily accessible electronic system. Similarly, the use of over-the-counter medications may have been underestimated, as these are not always documented in patients' medical records. Another potential limitation is that patients with comorbidities, such as cancer, rarely receive outpatient treatment. Additionally, costs for patients without complications could vary between institutions due to different protocols, especially at UNOP, where most patients are admitted even if they do not present complications. This variability could overestimate costs, although this was not reported. Currently, varicella is not a notifiable disease in Guatemala, and the country's annual rates caused by this disease were not found. The lack of these data contributes to the limitation of this study in estimating the disease burden accurately.

## Conclusions

Hospitalizations due to varicella in Guatemala are associated with considerable economic and clinical burden. Complications related to this disease are common and have a significant impact on the costs associated with medical care, especially in terms of surgery and ICU admission.

Varicella remains a major public health problem in Guatemala, affecting both healthy and immunocompromised children. The introduction of a universal varicella vaccination program has the potential to alleviate the clinical and economic burden associated with varicella in the country. In addition, the implementation of recommended control measures for household contacts and healthcare personnel can help prevent the spread of the disease.

## Author contributions

**Conceptualization:** Mario Melgar, Sebastian Medina.

**Data curation:** Ingrid Sajmolo, Ana Gabriela Cordova, Luis Hernández, Irwing Rivera, Ashly Zuñiga.

**Formal analysis:** Mario Melgar, Ingrid Sajmolo, André Chocó.

**Funding acquisition:** Mario Melgar.

**Investigation:** Mario Melgar, Ingrid Sajmolo.

**Methodology:** Ingrid Sajmolo, Sebastian Medina.

**Project administration:** Mario Melgar, Lidia Ortiz.

**Software:** André Chocó, Lidia Ortiz.

**Supervision:** Mario Melgar, Ingrid Sajmolo, Ashly Zuñiga, Claudia Beltrán, Sebastian Medina, Marcel Marcano-Lozada.

**Validation:** Ingrid Sajmolo, Sebastian Medina, Marcel Marcano-Lozada.

**Visualization:** Ingrid Sajmolo.

**Writing – original draft:** Mario Melgar, Ingrid Sajmolo, André Chocó.

**Writing – review & editing:** Mario Melgar, Ingrid Sajmolo, Sebastian Medina.

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
