## [Decision Letter · Decision Letter 0]

20 Aug 2024

PGPH-D-24-01383

Clinical and economic burden of varicella in pediatric patients hospitalized in four institutions in Guatemala.

Dear Dr. Sajmolo,

Thank you for submitting your manuscript to PLOS Global Public Health. After careful consideration, we feel that it has merit but does not fully meet PLOS Global Public Health’s publication criteria as it currently stands. Therefore, we invite you to submit a revised version of the manuscript that addresses the points raised during the review process.

Please revise your manuscript, paying particular attention to the comprehensive suggestions from Reviewer 2.

We look forward to receiving your revised manuscript.

Kind regards,

Jennifer Tucker, PhD

Staff Editor

Journal Requirements:

1. Please provide additional details regarding participant consent. If you are reporting a retrospective study of medical records or archived samples, please ensure that you have discussed whether the IRB or ethics committee waived the requirement for informed consent. If patients provided informed written consent to have data from their medical records used in research, please include this information.

2. In the online submission form, you indicated that "The data that support the findings of this study are available on request from the corresponding author". 

a. In a public repository, 

b. Within the manuscript itself, or 

c. Uploaded as supplementary information.

Additional Editor Comments (if provided):

Reviewers' comments:

Reviewer's Responses to Questions

**Comments to the Author**

1. Does this manuscript meet PLOS Global Public Health’s publication criteria ? Is the manuscript technically sound, and do the data support the conclusions? The manuscript must describe methodologically and ethically rigorous research with conclusions that are appropriately drawn based on the data presented.

Reviewer #1: Yes

Reviewer #2: Partly

2. Has the statistical analysis been performed appropriately and rigorously?

Reviewer #1: Yes

Reviewer #2: I don't know

3. Have the authors made all data underlying the findings in their manuscript fully available (please refer to the Data Availability Statement at the start of the manuscript PDF file)?

Reviewer #1: Yes

Reviewer #2: No

4. Is the manuscript presented in an intelligible fashion and written in standard English?

Reviewer #1: Yes

Reviewer #2: No

5. Review Comments to the Author

Reviewer #1: Although the authors have described this as a retrospective descriptive study, it would be useful to use some statistical parameter to confirm the difference between the group of patients with complications versus patients without complications, e.g. Chi-square test.

in the page 10, show: "while factors such as age, sex, ethnicity, immunocompromise, and outcome did not show clinically significant differences (p < 0.05)" Should be expressed as p>0.05.

Reviewer #2: Please see my attached review.

The only additional comments I have, other than outlined in the attachment, are that (1) dates when the site was accessed should be included in references for websites, and (2) if possible, a URL (and access date) should be provided for reference #4.

6. PLOS authors have the option to publish the peer review history of their article (what does this mean? ). If published, this will include your full peer review and any attached files.

**Do you want your identity to be public for this peer review?** For information about this choice, including consent withdrawal, please see our Privacy Policy .

Reviewer #1: **Yes: ** Antonio J García-Ruiz. Health Economics & Outcomes Research. Dep Pharmacology & Clinical Therapeutics. School of Medicine. University of Malaga

Reviewer #2: No

While revising your submission, please upload your figure files to the Preflight Analysis and Conversion Engine (PACE) digital diagnostic tool, https://pacev2.apexcovantage.com/ . PACE helps ensure that figures meet PLOS requirements. To use PACE, you must first register as a user. Registration is free. Then, login and navigate to the UPLOAD tab, where you will find detailed instructions on how to use the tool. If you encounter any issues or have any questions when using PACE, please email PLOS at figures@plos.org. Please note that Supporting Information files do not need this step.

---

## [Decision Letter · Decision Letter 1]

15 Dec 2024

PGPH-D-24-01383R1

Clinical and economic burden of varicella in pediatric patients hospitalized in four institutions in Guatemala.

Dear Dr. Sajmolo,

Thank you for submitting your manuscript to PLOS Global Public Health. After careful consideration, we feel that it has merit but does not fully meet PLOS Global Public Health’s publication criteria as it currently stands. Therefore, we invite you to submit a revised version of the manuscript that addresses the points raised during the review process. In particular, please check that all references are correct.

We look forward to receiving your revised manuscript.

Kind regards,

W. Alton Russell, PhD

Academic Editor

Journal Requirements:

Additional Editor Comments (if provided):

Reviewers' comments:

Reviewer's Responses to Questions

**Comments to the Author**

1. If the authors have adequately addressed your comments raised in a previous round of review and you feel that this manuscript is now acceptable for publication, you may indicate that here to bypass the “Comments to the Author” section, enter your conflict of interest statement in the “Confidential to Editor” section, and submit your "Accept" recommendation.

Reviewer #1: (No Response)

Reviewer #2: (No Response)

2. Does this manuscript meet PLOS Global Public Health’s publication criteria ? Is the manuscript technically sound, and do the data support the conclusions? The manuscript must describe methodologically and ethically rigorous research with conclusions that are appropriately drawn based on the data presented.

Reviewer #1: Yes

Reviewer #2: Yes

3. Has the statistical analysis been performed appropriately and rigorously?

Reviewer #1: Yes

Reviewer #2: I don't know

4. Have the authors made all data underlying the findings in their manuscript fully available (please refer to the Data Availability Statement at the start of the manuscript PDF file)?

Reviewer #1: No

Reviewer #2: Yes

5. Is the manuscript presented in an intelligible fashion and written in standard English?

Reviewer #1: Yes

Reviewer #2: Yes

6. Review Comments to the Author

Reviewer #1: Citation number 4 is still missing the correct reference.

As well as quotes 8 and 9. The access day is missing.

Reviewer #2: This updated version of the manuscript is much tighter and cleaner than the previous version -- well done! I have only a few minor comments remaining:

1. Materials and Methods, 1st paragraph, last sentence -- "... and where hospitalized" -- Please change "where" to "who were".

2. Results, 3rd paragraph below Table 4, 3rd sentence -- "They also had a higher number of pa specialist consutations..." -- what is "pa"?

3. Overall -- Please double check all citation numbers. The citation number for Giglio in the final paragraph of the Discussion is incorrect; I have not confirmed the rest of the citations.

7. PLOS authors have the option to publish the peer review history of their article (what does this mean? ). If published, this will include your full peer review and any attached files.

**Do you want your identity to be public for this peer review?** For information about this choice, including consent withdrawal, please see our Privacy Policy .

Reviewer #1: **Yes: ** Antonio J Garcia-Ruiz, MD, PhD,MsC. Health Economics & Outcomes Research. University of Málaga (Spain)

Reviewer #2: No

While revising your submission, please upload your figure files to the Preflight Analysis and Conversion Engine (PACE) digital diagnostic tool, https://pacev2.apexcovantage.com/ . PACE helps ensure that figures meet PLOS requirements. To use PACE, you must first register as a user. Registration is free. Then, login and navigate to the UPLOAD tab, where you will find detailed instructions on how to use the tool. If you encounter any issues or have any questions when using PACE, please email PLOS at figures@plos.org. Please note that Supporting Information files do not need this step.

---

## [Editor Report · Decision Letter 2]

24 Dec 2024

Clinical and economic burden of varicella in pediatric patients hospitalized in four institutions in Guatemala.

PGPH-D-24-01383R2

Dear M.D. Sajmolo,

We are pleased to inform you that your manuscript 'Clinical and economic burden of varicella in pediatric patients hospitalized in four institutions in Guatemala.' has been provisionally accepted for publication in PLOS Global Public Health.

Best regards,

W. Alton Russell, PhD

Academic Editor